# ACCURACY-PRIVACY TRADE-OFF IN DEEP ENSEMBLE: A MEMBERSHIP INFERENCE PERSPECTIVE

## ABSTRACT

Deep ensemble learning has been shown to improve accuracy by training multiple neural networks and fusing their outputs. Ensemble learning has also been used to defend against membership inference attacks that undermine privacy. In this paper, we empirically demonstrate a trade-off between these two goals, namely accuracy and privacy (in terms of membership inference attacks), in deep ensembles. Using a wide range of datasets and model architectures, we show that the effectiveness of membership inference attacks also increases when ensembling improves accuracy. To better understand this trade-off, we study the impact of various factors such as prediction confidence and agreement between models that constitute the ensemble. Finally, we evaluate defenses against membership inference attacks based on regularization and differential privacy. We show that while these defenses can mitigate the effectiveness of the membership inference attack, they simultaneously degrade ensemble accuracy. We illustrate similar trade-off in more advanced and state-of-the-art ensembling techniques, such as snapshot ensembles and diversified ensemble networks. The source code is available in supplementary materials.

## 1 INTRODUCTION

Ensemble learning has been shown to improve classification accuracy of neural networks in particular, and machine learning classifiers in general (Kondratyuk et al., 2020; Kuncheva & Whitaker, 2003; Sagi & Rokach, 2018). The most commonly used approach for deep models involves averaging the output of multiple neural networks (NN) that are independently trained on the same dataset with different random initialization, called **deep ensemble** (Lobacheva et al., 2020). Such a simple approach has been extensively used in practice to improve accuracy (Lee et al., 2015; Wang et al., 2020). Notably, a majority of the top performers in machine learning benchmarks, such as the ImageNet Large Scale Visual Recognition Challenge (Russakovsky et al., 2015), have adopted some form of ensemble learning (Lee et al., 2015; Szegedy et al., 2015; He et al., 2016).

Other forms of ensemble learning (different from deep ensembles), such as partitioning, has also been used to defend against privacy-harming membership inference (MI) attacks, where the goal of an attacker is to infer whether a sample has been used to train a model–i.e., whether the sample belongs to the train set. Membership inference attacks generally use the prediction confidence of NN models to infer membership status of a sample (Salem et al., 2018; Shokri et al., 2017; Truex et al., 2019; Yeom et al., 2018) by leveraging the insight that trained models may output higher prediction confidence on train samples than non-train samples (Choo et al., 2020). The intuition behind using ensemble learning approaches, like partitioning, to defend against MI attacks is that training each model on a different subset of data makes the ensemble less prone to overfitting (Salem et al., 2018). This idea of using ensemble learning to defend against MI attacks has since been discussed in the literature (Huang et al., 2020; Li et al., 2021; Rahimian et al., 2020; Yang et al., 2020). However, none of these papers theoretically or empirically demonstrate usefulness of ensemble learning as a defense mechanism.

In this paper, we show that these two goals of ensemble learning, namely improving accuracy and defending against MI attack, do not trivially sum up in a unified solution. Figure 1 illustrates accuracy and privacy trade-off by plotting accuracy and membership inference attack effectiveness for ensembles comprising of varying number of base models (1, 2, 5, and 10) that are trained for different numbers of epochs (5, 45, and 90). We make two key observations here. First, there is an increase in

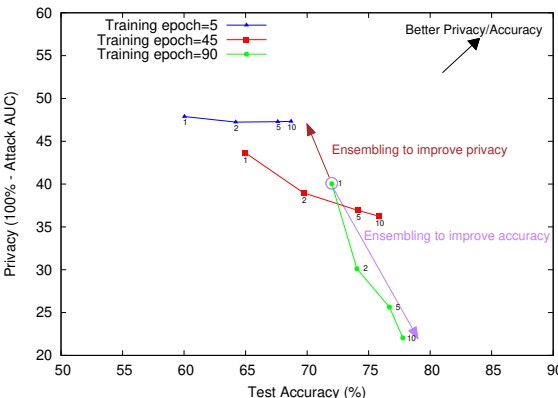

Figure 1: Trade-off between accuracy and privacy on an AlexNet model trained on CIFAR10. Each curve contains four points corresponding to ensembles comprising of 1, 2, 5, and 10 base models (from left to right). Using the single model trained for 90 epochs as a baseline, there are two choices: (1) making an ensemble of these models to achieve the highest accuracy possible but worse privacy (purple arrow), or (2) making an ensemble of less overfitted models (epoch #5) to achieve slightly lower accuracy of a single model but better privacy (brown arrow).

both accuracy and MI attack effectiveness as we go from a single model to ensembles comprising of an increasing number of base models. The trade-off is more noticeable for more accurate models trained for a larger number of epochs. Second, we can adapt the design of ensembles to suitably navigate the trade-off between accuracy and privacy. Specifically, the two extreme cases are to: (1) maximize accuracy by using an ensemble of highly accurate models but at the cost of worse privacy[1] (purple arrow); and (2) maximize privacy by intentionally using an ensemble of *under-fitted* models instead of a single model but at the cost of accuracy (brown arrow).

To understand the root cause of this trade-off, we show that using deep ensembles to improve accuracy exacerbates its susceptibility to membership inference attacks by making train and non-train samples more distinguishable. By analyzing the confidence averaging mechanism of deep ensembles, we investigate potential factors that enable membership inference. We show that the most influential factor is the level of correct agreement among models. Simply put, the number of models that correctly classify a train sample is often greater than the ones that correctly classify a test sample. This results in a wider confidence gap between train and non-train samples, when confidence values are averaged, enabling more effective membership inference attacks.

We further show that the difference in the level of correct agreement between train and non-train samples is correlated with models' generalization gap. Hence, a natural question to ask is "can deep ensembles that use less overfitted models mitigate privacy issues while achieving high accuracy?". To answer this question, we study several regularization techniques, common membership inference defenses, and a few other ensembling approaches. We again observe a privacy-accuracy trade-off pattern similar to that shown in Figure 1.

*Summary of contributions:* In this paper, we perform a systematic empirical study of MI attacks on deep ensemble models. First, we show that when deep ensembles improve accuracy, it also leads to a different distribution shift in the prediction confidence of train and test samples, which in turn enables more effective membership inference. Second, we analyze various factors that potentially cause the prediction confidence of train and non-train samples to diverge. Among potential factors, we show that the most dominant factor is the level of correct agreement among models which indicates that more models in an ensemble agree on their prediction when a sample is a training sample. Hence, the aggregation of their prediction yields higher confidence output in comparison with non-train samples. We show that common defense mechanisms in membership inference literature, including differential privacy, MMD+Mixup, L1 and L2 regularization, as well as other ensemble training approaches, such as bagging, partitioning, and stacking (Salem et al., 2018), can be used to mitigate

---

[1]Note that for complicated tasks, such as image classification, the common practice is to train deep models for a large number of epochs and avoid under-fitted models. That is because memorizing samples from long-tailed subpopulations are shown to be necessary to achieve close-to-optimal generalization error (Feldman, 2020).

effectiveness of MI attacks but at the cost of accuracy. Although the main focus of the paper is on deep ensembles, we also cover bagging, partitioning, stacking (Salem et al., 2018), logit averaging (Appendix A.3), weighted averaging (Appendix A.4), as well as more advanced and state-of-the-art ensembling techniques, such as snapshot ensembles (Huang et al., 2017) and diversified ensemble networks (Zhang et al., 2020) (Appendix A.5). We observe similar trade-off.

## 2 System Model

### 2.1 Ensemble Learning

**Background.** In literature, ensemble learning refers to various approaches that combine multiple models to make a prediction. Models used to construct an ensemble are often called base learners. There are two main factors to construct an ensemble (Sagi & Rokach, 2018): 1) how base learners are trained to ensure diversity, such as random initialization, bagging, partitioning, etc., and 2) how the output of base learners are fused to obtain the final output, including majority voting, confidence averaging, stacking, etc. Unlike ensemble of traditional machine learning algorithms, in a **deep ensemble**, the main source of diversity often comes only from random initialization of base learners (Fort et al., 2019). In fact, other sources of diversity, such as bagging, have been shown to considerably degrade the overall accuracy of a deep ensemble (Lee et al., 2015; Lakshminarayanan et al., 2017).

**System Model.** We mainly focus on the most widely used deep ensemble (Kondratyuk et al., 2020) unless otherwise specified. In this model, 1) base models are trained with random initialization on the same training dataset, and 2) their prediction confidence are fused through averaging. A less common approach is to average model logits which has been used in a few studies (Webb et al., 2020; Wang et al., 2020). See Appendix A.3 for experimental evaluation of logit averaging and A.4 for weighted averaging ensembles. We also evaluate two state-of-the-art deep ensembling approaches, namely snapshot ensemble and diversified ensemble network, in Appendix A.5. Other general ensembling approaches, such as bagging, partitioning, and stacking (Salem et al., 2018), are studied as defense mechanisms because they degrade accuracy but improve protection against MI attacks.

### 2.2 Membership Inference

**Background.** Membership inference is a form of privacy leakage where the goal is to determine if a sample was used during the training of a target model. Samples used during training are often referred to as member or train samples, and other samples are referred to as non-member, non-train, or test samples. The first MI attack on neural networks was proposed in Shokri et al. (2017) where the attacker trains an attack classifier to predict the membership status. The attack classifier takes the prediction confidence of a target model as an input. Assuming that the attacker has access to a dataset with a similar distribution, she trains a set of shadow models to mimic the target model. Since the membership status of the data with which the shadow models are trained are known to the attacker, she can use the data to train the attack classifier. Many papers use the same idea with different variations or less restrictive assumptions (Salem et al., 2018; Liu et al., 2019; Song et al., 2019; Long et al., 2017; Truex et al., 2019; Long et al., 2018; Yeom et al., 2018; Rezaei & Liu, 2021; Zou et al., 2020; Li & Zhang, 2020). Most previous work built upon the idea of using prediction confidence to infer the membership status, except for Rezaei & Liu (2021); Choo et al. (2020); Rahimian et al. (2020). In Rezaei & Liu (2021), the authors assumed white-box access to the target model and launched a series of MI attack based on confidence values, distance to the decision boundary, gradient w.r.t model weight, and gradient w.r.t input. In Choo et al. (2020), the authors proposed two attacks based on input transformation and distance to the boundary in a black-box setting. Similarly, in Rahimian et al. (2020), the attacker randomly perturbs an input to obtain a set of random transformations of the input and uses the predicted labels to infer membership status.

**System Model.** Since most existing attacks use confidence values, we first focus on changes of confidence values when using deep ensembles. We show that when using deep ensembles the distribution of confidence values becomes more distinguishable between train and non-train set in comparison with non-ensemble case. Consequently, any MI attack that relies on confidence values would be more effective on deep ensembles. Since our goal is to show a trade-off between accuracy and privacy, not to show which confidence-based attack can slightly outperforms another, we focus on a confidence-based attack proposed in Rezaei & Liu (2021) in both white-box and black-box

settings. Here, white-box means the attacker has access to base-learners' output before aggregation, and black-box means the attacker has only access to the aggregated confidence output. Decision boundary-based attacks are extremely computational and query inefficient and it is not trivial how to adopt them for ensemble learning where essentially an input is copied $n$ times and then fed to $n$ models. Gradient-based approach of Rezaei & Liu (2021) also needs full knowledge of the entire deep ensemble. We consider adaptation of non-confidence-based attacks for deep ensembles as future work.

### 2.3 MEMBERSHIP INFERENCE DEFENSES

**Background.** Defense mechanisms against membership inference attacks can be summarized into two categories Rahimian et al. (2020):

*Generalization-based:* Shokri (Shokri et al., 2017) was first to correlate membership inference success with overfitting. Since then, many standard regularization techniques have been used to alleviate overfitting, such as L1 regularization (Choo et al., 2020), L2 regularization (Choo et al., 2020; Truex et al., 2019; Nasr et al., 2019; Jia et al., 2019; Shokri et al., 2017), differential privacy (Choo et al., 2020; Rahimian et al., 2020), dropout (Jia et al., 2019), and adversarial training (Nasr et al., 2018). Interestingly, ensemble learning has also been proposed as a defense mechanism. In Salem et al. (2018), they proposed a combination of partitioning and stacking as a defense mechanism. The intuition is that training each model with different subset of data makes the entire ensemble model less prone to overfitting. Note that these defense mechanisms often degrade the accuracy of the target model (see Section 4.2) (Choo et al., 2020).

*Confidence-masking:* These defense mechanisms aim to reduce the amount of information that can be obtained from the output of a target model by perturbing (Jia et al., 2019) or limiting the dimensionality of the output (Shokri et al., 2017; Truex et al., 2019; Choo et al., 2020). Majority of these approaches are post-training methods and only work in black-box setting.

**System Model.** In this paper, we mainly focus on generalization-based defense mechanisms with the exception of MMD+Mixup (Li et al., 2021). This is because most confidence-masking approaches manipulate confidence values post-training. As a result, the output values of these models do not reliably represent the "confidence" of the model. These approaches are built under the assumption that accurate prediction of confidence is not needed. However, many applications require accurate estimation of confidence. Moreover, if the accurate prediction of confidence is not required, then the trivial MI defense would be to only output the class label and avoid these confidence-masking defenses altogether.

## 3 HOW DOES ENSEMBLING INCREASE MEMBERSHIP INFERENCE EFFECTIVENESS?

### 3.1 CONFIDENCE DISTRIBUTION SHIFT

Ensemble learning is only helpful when base learners disagree on some samples (Kuncheva & Whitaker, 2003; Sagi & Rokach, 2018). Otherwise, ensembling does not improve accuracy. Furthermore, when deep ensemble is used, the confidence values of multiple models are averaged to obtain the final prediction. Consequently, when ensembling improves accuracy, it averages the prediction confidence of highly confident predictions (mostly from models which correctly classified the sample) and less confident predictions (mostly from models which misclassified the sample). As a result, confidence distribution shift is inevitable for both train and test samples. This phenomenon can be observed as the distribution of Figure 2(a) shifts to that of 2(d). This can be better observed by separating correctly classified samples which have significantly higher prediction confidence (Figure 2(e)) and misclassified samples which have lower prediction confidence (Figure 2(f)). One can clearly observe that both distributions shift more towards the center from Figure 2(b) and (c) to Figure 2(e) and (f). However, the change in the distribution of train and test samples does not necessarily cause a more effective membership inference if the change has a similar effect on the confidence distribution of both train and test samples. In the next subsection, we analyze the potential factors that affect the distribution change and how they can change confidence distribution of member and non-member sets differently.

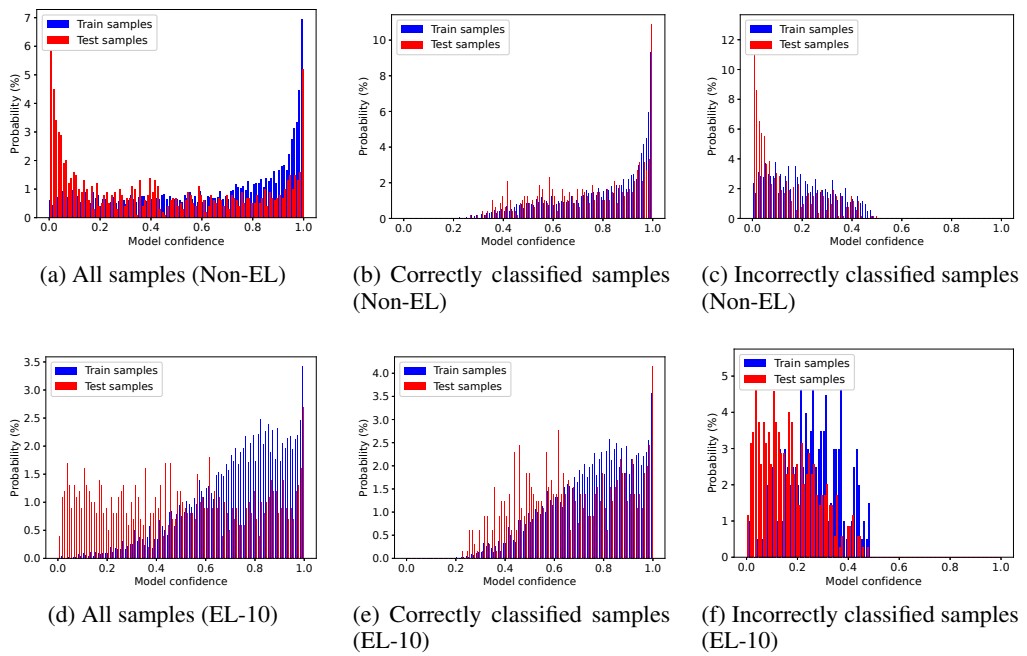

Figure 2: AlexNet on CIFAR10 (Class #2): Confidence distribution of a single model (top row) versus an ensemble of 10 models (bottom row). Jensen–Shannon divergence of the two distributions are as follows: a) .2777, b) .1595, c) .2850, d) .3575, e) .2134, and f) .3792.

## 3.2 EFFECT OF ENSEMBLING ON INDIVIDUAL SAMPLES

We use $y_i$ to denote the confidence value of the $i^{th}$ model in an ensemble of $n$ models. Hence, for a given sample $x$, the output of the ensemble is:

$$y_{el}(x) = \frac{\sum^n y_i(x)}{n} = \frac{\sum^c y_i^c(x) + \sum^m y_i^m(x)}{n}. \tag{1}$$

Given a single sample, we can further divide models in the ensemble into two groups: 1) models that correctly classified the sample denoted by $y_i^c$ and 2) models that misclassified the sample by $y_i^m$. For a given sample $x$, $c$ models correctly classify it and $m$ models misclassify, where $c + m = n$. Note that the value of $c$ and $m$ depends on the sample[2].

Based on the Eq. (1), three major factors affect the final confidence value ($y_{el}$) of a sample: $y^c$, $y^m$, and $c$. As a result, if these values are different for train and test samples, the ensembling causes different shift in the distributions, and consequently, membership inference attack will be more effective. These factors are as follows:

1. Confidence value of correctly classifying models ($y^c$): Since the majority of samples are supposed to be correctly classified by a practical model, any distinguishable confidence difference between train and test samples can lead to a very effective membership inference attack. However, as shown in Figure 2(b), we can observe that there is no significant difference between train and test samples.

2. Confidence value of misclassifying models ($y^m$): Unlike correctly classified samples, there is a marginal difference between confidence distribution of train and test samples of misclassified samples (see Figure 2(c)). This may be exploited by membership inference attack to partially distinguish between train and test samples.

3. Level of correct agreement ($c$) among models: As shown in Figure 3(a), the number of models that correctly classify a sample ($c$) is greater for train samples than test samples (see

---

[2]By an abuse of notation, we use $c$ ($m$) to refer to (in)correctly classifying models and also as a superscript for the model output of (in)correctly classifying models, that is, $y_i^c$ ($y_i^m$).

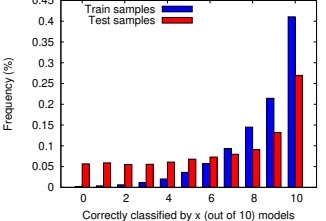
(a) Correct agreement distribution

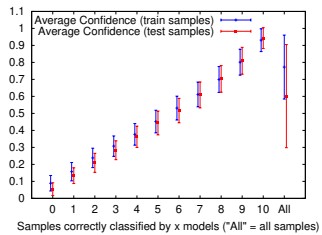
(b) Distribution difference at each agreement level

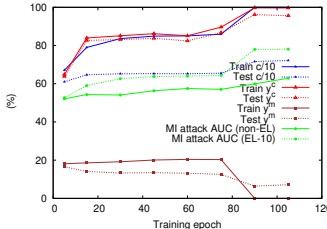
(c) The effect of three factors on MI attack

Figure 3: AlexNet model trained on CIFAR10. Left: The distribution of the number of times a sample is correctly classified by 10 models used in the ensemble. The models often make less classification mistake on train samples than test samples. Center: By separating samples based on how many times they have been correctly classified, we can observe that the confidence output of these samples are negligible between train and test sets. Only when all samples are compared the distribution difference is significant and that is the direct effect of the third factor, namely the level of correct agreement. Right: The effect of the three factors on the MI attack. As the gap between the level of correct agreement of train and test widens (the blue lines), the MI attack on ensembles becomes more effective than a single model (green lines).

Appendix A.2 for more results). Since prediction confidence of correctly classified samples are higher than misclassified samples on average, i.e., $y^c > y^m$, and $c$ is smaller for test samples, the ensemble confidence of test samples ($y_{el}$) becomes lower than train samples. As a result, this factor can largely contribute to the effectiveness of membership inference attacks on deep ensembles.

We note that the first two factors are not unique to ensembles and can be exploited by an attacker in a single model scenario as well. As a result, these two factors have been extensively studied in Rezaei & Liu (2021) in a single model scenario across various image datasets and well-trained models. They have shown that for deep models the first factor ($y^c$) is almost indistinguishable between train and test set and only the second factor ($y^m$) is marginally distinguishable. However, this marginal difference does not have a considerable impact on the different distribution shift in train and test sets.

On the other hand, **the level of agreement** has a big impact on different distribution shifts. To better demonstrate this, we can analyze the distribution difference in each level of agreement separately. As shown in Figure 3(b), the average confidence of train and test samples are very close and indistinguishable when each level of agreement is drawn separately. If the effect of the first two factors were considerable, the two confidence values for each level of agreement would have been more distinguishable. Note that the average confidence between train and test sets is more distinguishable for the first two points in x-axis (where majority of models misclassify a sample). However, these distributions only constitute a tiny portion of the training dataset, as shown with the first two blue bins in Figure 3(a). However, when all samples are combined, we can vividly observe that the average confidence gap between train and test sets considerably widens, as shown in the last point in x-axis in Figure 3(b). This clearly demonstrates that the major factor in different distribution shift between train and test sets is the level of agreement ($c$).

Note that, unlike the first two factors, the third factor ($c$) only improves the effectiveness of membership inference attacks in ensemble scenario because it does not exist in a single model. In other words, if a defence strategy eliminates the effect of the average level of correct agreement (i.e., it ensures that $c$ is close between train and test samples), the membership inference attack is still possible on the ensemble, but only to the degree that it is possible on a single model[3]. As shown in Figure 3(c), as the gap between $y^c$ of train and test sets (red lines) and the gap between $y^m$ of train and test sets (brown lines) increases, the attack on both single model (non-ensemble) and also the ensemble

---

[3]Although this can be understood by analyzing the Eq. (1), it is difficult to demonstrate empirically. The reason is that these three factors are not independent, and, hence, our attempts to significantly change the third factor without changing the other two factors have been unsuccessful. See Appendix A.2 to see the dependency between these two factors across all datasets and models.

(EL-10) increases. However, only when the average level of correct agreement gap between train and test (blue lines) widens, the membership inference attack on ensembles becomes more effective than on non-ensembles.

Another important observation from Figure 3(c) is that the minimum level of agreement gap between train and test occurs when models are relatively underfitted (i.e., the blue lines in first few epochs). This phenomena has also been partially observed in Fort et al. (2019) (Figure 2(c)). The main reason is that underfitted models often only learn the most common and generalizable features and, thus, they often agree on the features and predictions. As they move from underfitted region to overfitted region, their generalization gaps widen (blue lines in Figure 3(c)). As a result, they tend to correctly classify train samples more often than test samples. Consequently, they agree on train samples more than test samples and, hence, average gap of correct agreement between train and test set widens. Hence, the wider generalization gap of base learners is, the more effective membership inference attack would be on deep ensembles.

### 3.3 Why Does it Outperform Gap Attack Significantly?

Recently, several studies report a simple baseline attack called *gap attack* (Choo et al., 2020), also known as *naive attack* (Rezaei & Liu, 2021; Leino & Fredrikson, 2020) that achieves similar performance as the confidence-based attacks in most scenarios. The gap attack predicts a sample as member if it is correctly classified by the target model, and as non-member otherwise. In other words, gap attack essentially reflects the generalization gap (Rezaei & Liu, 2021). In Rezaei & Liu (2021), authors extensively analyzed this phenomena in deep models and argued current MI attacks that barely outperform gap attacks are ineffective in practice because they only reflect the generalization gap and cannot infer the membership status of each individual sample accurately.

Figure 4 shows that the effectiveness of membership inference attacks increases and outperforms the gap attack as deep ensembles are deployed. This raises significant privacy concern since the gap attack is often suggested as a baseline that is also hard to outperform in non-ensemble setting (Rezaei & Liu, 2021; Leino & Fredrikson, 2020; Choo et al., 2020). Note that gap attack can be viewed as a metric directly reflecting the generalization gap rather than a reliable membership inference. As suggested in Rezaei & Liu (2021), we can separate correctly classified samples and misclassified samples to understand why membership inference attacks can barely outperform gap attack. As shown in Figure 2(b) and (c), the distributions of train and test samples are similar when separated into correctly classified and misclassified sets. The reason why the distribution of all samples (Figure 2(a)) looks more distinguishable when correctly classified samples and misclassified samples are combined is that there are often more misclassified samples in the test set than the train set. This is the information that gap attack exploits which essentially reflects the generalization gap. In order for a membership inference attack to considerably outperform the gap attack, the distribution of correctly classified samples and misclassified samples should leak membership status information, which is not often the case as it is shown in Figure 2(b) and (c), and Rezaei & Liu (2021).

When ensembling is used, the distribution of confidence values changes dramatically, as explained in Section 3.1. By comparing the confidence distribution of correctly classified samples in an ensemble (Figure 2(e)) with a non-ensemble scenario (Figure 2(b)), the distribution is clearly more distinguishable in ensemble case. This is of significant privacy concern because, as discussed in Rezaei & Liu (2021), majority of samples in practice belong to the correctly classified set. Similar trend is also observable in misclassified samples (Figure 2(f)). Hence, the confidence values, that barely leak more information than generalization gap itself in a single model scenario, now considerably leak more membership information than just the generalization gap. That is the reason why membership inference attacks are significantly more effective in deep ensembles in comparison to the gap attack.

## 4 Experiments Results

### 4.1 Experimental Setup

We explore a wide range of datasets that are often used in deep ensemble literature or membership inference literature: Adult[4], Texas, Purchase (Shokri et al., 2017), MNIST (LeCun et al., 1998),

---

[4]http://archive.ics.uci.edu/ml/datasets/Adult

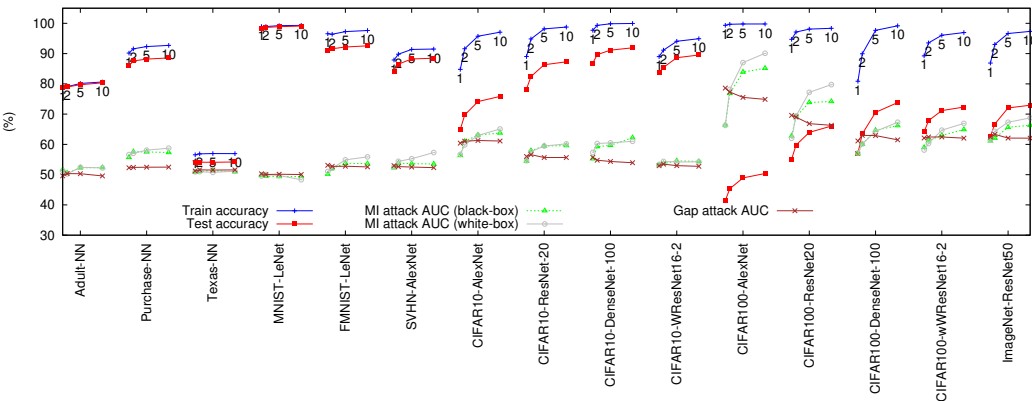

Figure 4: Membership inference attack results across all datasets/models. Each curve indicates an ensemble of 1 (non-EL), 2, 5, and 10 models from left to right.

FMNIST (Xiao et al., 2017), SVHN (Netzer et al., 2011), CIFAR10 (Krizhevsky et al., 2009), CIFAR100 (Krizhevsky et al., 2009), and ImageNet (Russakovsky et al., 2015). For non-image datasets (Adult, Purchase, and Texas), we use a fully connected neural network consisting of a hidden layer of size 128 and a Softmax layer. All other training parameters for these datasets are set as suggested in Shokri et al. (2017). For image datasets, we use a wide range of convolutional neural networks depending on the input dimension and the difficulty of the task. We use the model implementations adopted in Nasr et al. (2019); Rezaei & Liu (2021)[5]. We train 10 models for each dataset with random initialization and construct an ensemble of 2, 5 and 10 models, respectively.

For membership inference attack models, we consider two different threat models: black-box and white-box. In the black-box setting, the attacker only has access to the output of the ensemble, which is the average over softmax output of individual models. In the white-box setting, the attacker can see the output of each model in ensemble. In the former case, the attack model takes the output of ensemble as input, while in the latter case, it takes all confidence outputs of all models as input. Attack models are NNs with three hidden layers of size 128, 128, and 64, respectively. We consider a worst case scenario where $80\%$ of the training dataset is given to the attacker and the goal is to infer the membership of the remaining samples, similar to Rezaei & Liu (2021). All other training parameters are set as suggested in Rezaei & Liu (2021). All experiments are implemented in Python 3.5, PyTorch 1.6 and are conducted on a server with Intel Xeon Processor 24-core 2.70GHz, 8 GeForce RTX 2080 GPUs and 384GB memory. See Appendix A.1 for full report. The results of the logit averaging, weighted averaging, and snapshot ensembles and diversified ensemble networks are shown in Appendix A.3, Appendix A.4, and Appendix A.5, respectively.

Figure 4 shows the results on all datasets. For some datasets, such as Adult, Texas, and MNIST, ensemble learning barely changes the accuracy or privacy. That is because the disagreement across models is insignificant in these datasets. For all other datasets, ensemble learning improves the accuracy as well as the effectiveness of membership inference attacks. As mentioned in Section 3.2, the most salient factor in membership inference effectiveness on deep ensembles is the accuracy gap between train and test set. Figure 4 clearly shows that whenever this generalization gap is large for non-ensemble case, the attack improvement is significant after ensembling. It is worth noting that the ensembling can often reduce the generalization gap and the effectiveness of the gap attack (e.g., CIFAR10-DenseNet-100, CIFAR100-AlexNet, or CIFAR100-ResNet20). However, due to the reasons explained in Section 3.3, the membership inference effectiveness still increases.

## 4.2 DEFENSE MECHANISMS

As discussed in previous sections, the main factor that causes more privacy leakage in ensemble learning is the generalization gap of the base models. Therefore, any standard regularization technique can potentially work as a defense mechanism. In this paper, we study L1 and L2 regularization,

---

[5]https://github.com/bearpaw/pytorch-classification

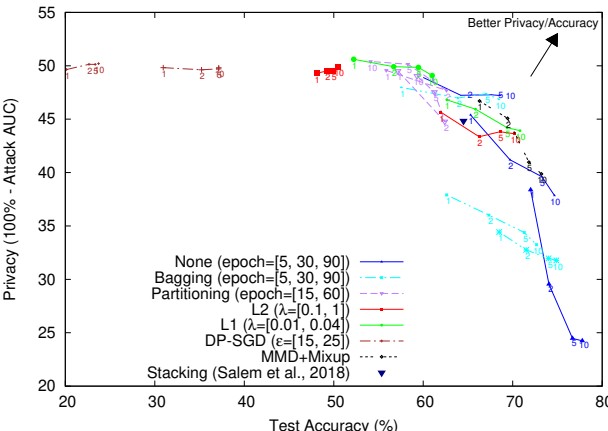

Figure 5: Effect of defense mechanisms on an AlexNet model trained on CIFAR10. The size of each point indicates the relative value of its parameter, i.e., epoch, $\lambda$ and $\epsilon$.

DP-SGD[6], and MMD+Mixup (Li et al., 2021). Furthermore, one can simply terminate training early to keep the model's weights in less overfitted region. Moreover, some ensembling techniques, such as bagging, partitioning, and stacking (Salem et al., 2018), limits the access of models to all training samples, which can potentially reduce membership inference effectiveness on deep ensembles.

We train 10 AlexNet models with different initialization on CIFAR10 for each defense mechanism. Here, we consider black-box membership inference attack to measure privacy. Figure 5 shows the effect of defense mechanisms on ensemble learning. We can observe a consistent trade-off between ensemble accuracy and privacy that resembles Pareto optimal points. Note that privacy degradation rate is clearly not constant. An ensemble of heavily regularized models or under-fitted models barely causes more privacy leakage (e.g., L2 regularization with $\lambda = 1$). On the other hand, an ensemble of overfitted models (e.g., non-regularized models trained for 90 epochs) results in large privacy leakage. It is worth mentioning that for a given accuracy, some approaches provide more privacy. For instance, the regular deep ensemble and the bagging of models trained for 30 epochs achieve similar accuracy. However, bagging is in fact more prone to membership inference because each model in bagging is exposed to fewer training data samples and consequently is more overfitted.

## 5 Conclusion & Discussion

In this paper, we investigate membership inference attacks in deep ensemble learning and demonstrate that there is a trade-off between accuracy and privacy. We note some limitations of our empirical analysis and opportunities for future work. First, privacy is a multi-faceted concept and can be defined or quantified in several ways. In this work, we quantified privacy leakage in terms of the effectiveness of membership inference attacks. Future work can quantify privacy in terms of other relevant attacks such as model inversion (Fredrikson et al., 2015; He et al., 2019), property inference (Ateniese et al., 2015; Ganju et al., 2018), and model stealing attack (Tramèr et al., 2016) as well as formally-provable measures such as differential privacy (Dwork, 2008). Second, ensemble learning is an umbrella term covering a variety of methods to combine multiple base learners. An arbitrary method of training and combining base learners can be construed as ensemble learning. We mainly focused to ensemble averaging because of its prevalent use for ensembling deep models. Given the privacy leakage of deep ensembles demonstrated here, future work can look into designing other fusion approaches to better navigate the trade-off between privacy and accuracy. One solution can focus on training models sequentially and applying some non-trivial criteria during training of each model to force the distribution of correct agreement to be close for train and non-train samples. How to achieve this is not trivial and needs further research.

---

[6]We use Opacus implementation in PyTorch from https://github.com/pytorch/opacus.

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

## A    APPENDIX

### A.1    FULL TRAINING REPORT

Figure 6 demonstrates the improvement of accuracy and MI attack over various training epochs. For datasets that ensembling outperforms a single model, using an ensemble of underfitted models is less prone to MI attack. However, it corresponds to lower accuracy. This is consistent with all experiments across different datasets and models, shown in Figure 6.

### A.2    LEVEL OF CORRECT AGREEMENT

As discussed in section 3.2, overfitted models tend to disagree more on test samples than train samples. In other word, the distribution of agreement for train and test sets becomes more distinguishable as models overfit. This distinction is more clear for datasets, such as CIFAR10 and CIFAR100, which shows most improvement when ensembling is used, as shown in Figure 7, 8, and 9. Furthermore, the level of agreement can reveal if an ensemble can actually improve prediction. If all models correctly classify a sample or all models misclassify a sample, ensembling fails to outperform a single model. This is the case for Adult, Texas, and MNIST, as shown in Figure 9.

### A.3    AVERAGING LOGIT

In this section, we average logits (the output of a model before Softmax) of NN models instead of the confidences. We can still observe that ensembling leaks more membership status than non-ensemble scenario. However, the MI attacks with average confidence (Figure 6), in general, are slightly more effective than MI attacks with average logits (Figure 10). The reason is that that confidence values are normalized and, hence, when aggregated all models have the same contribution to the overall confidence output of the ensemble. However, when logit is used, the confidence output of the ensemble is more influenced by highly activated neurons. These highly activated neurons, which

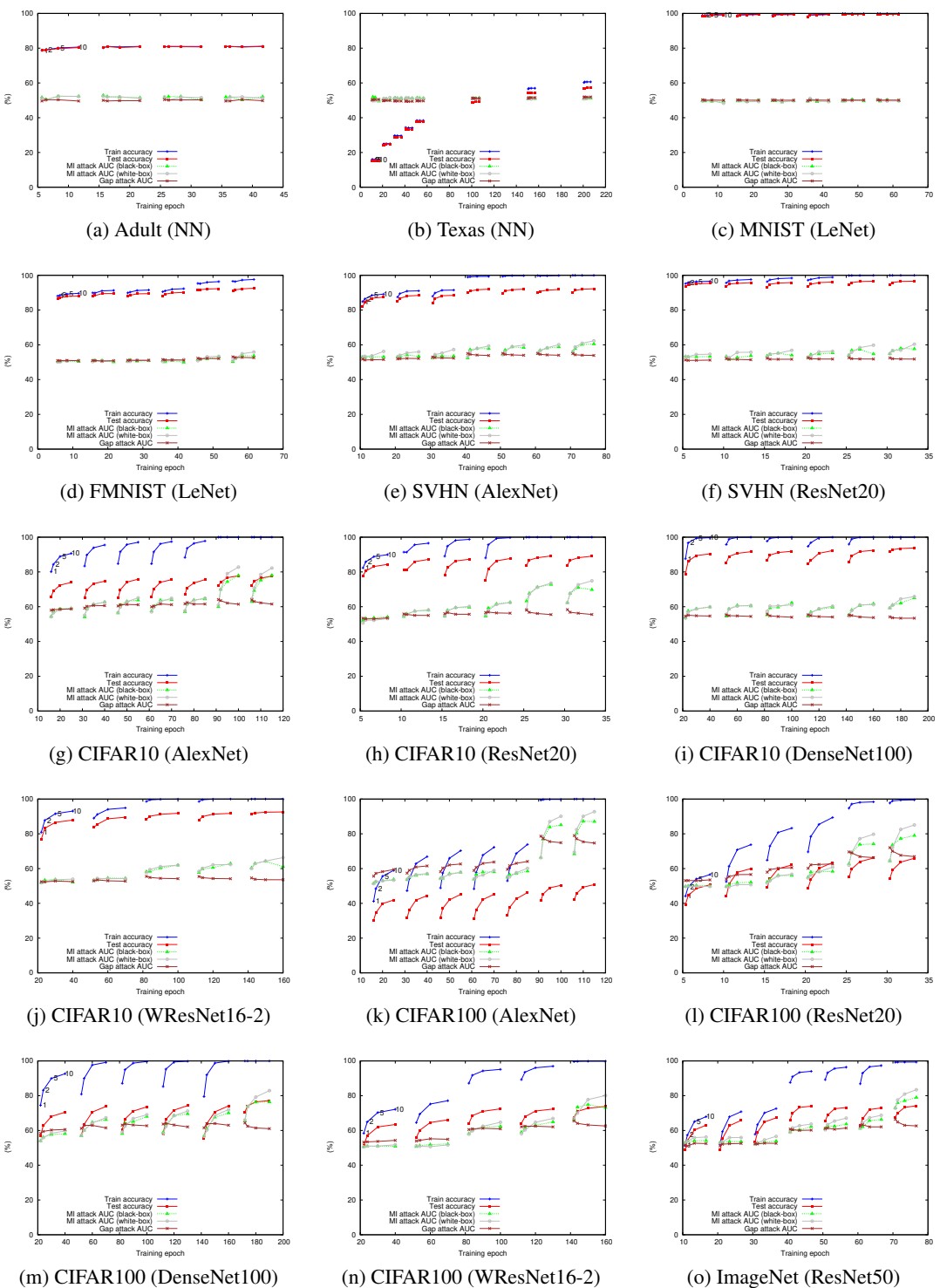

Figure 6: Target models' accuracy and MI attacks' AUC across all datasets and models.

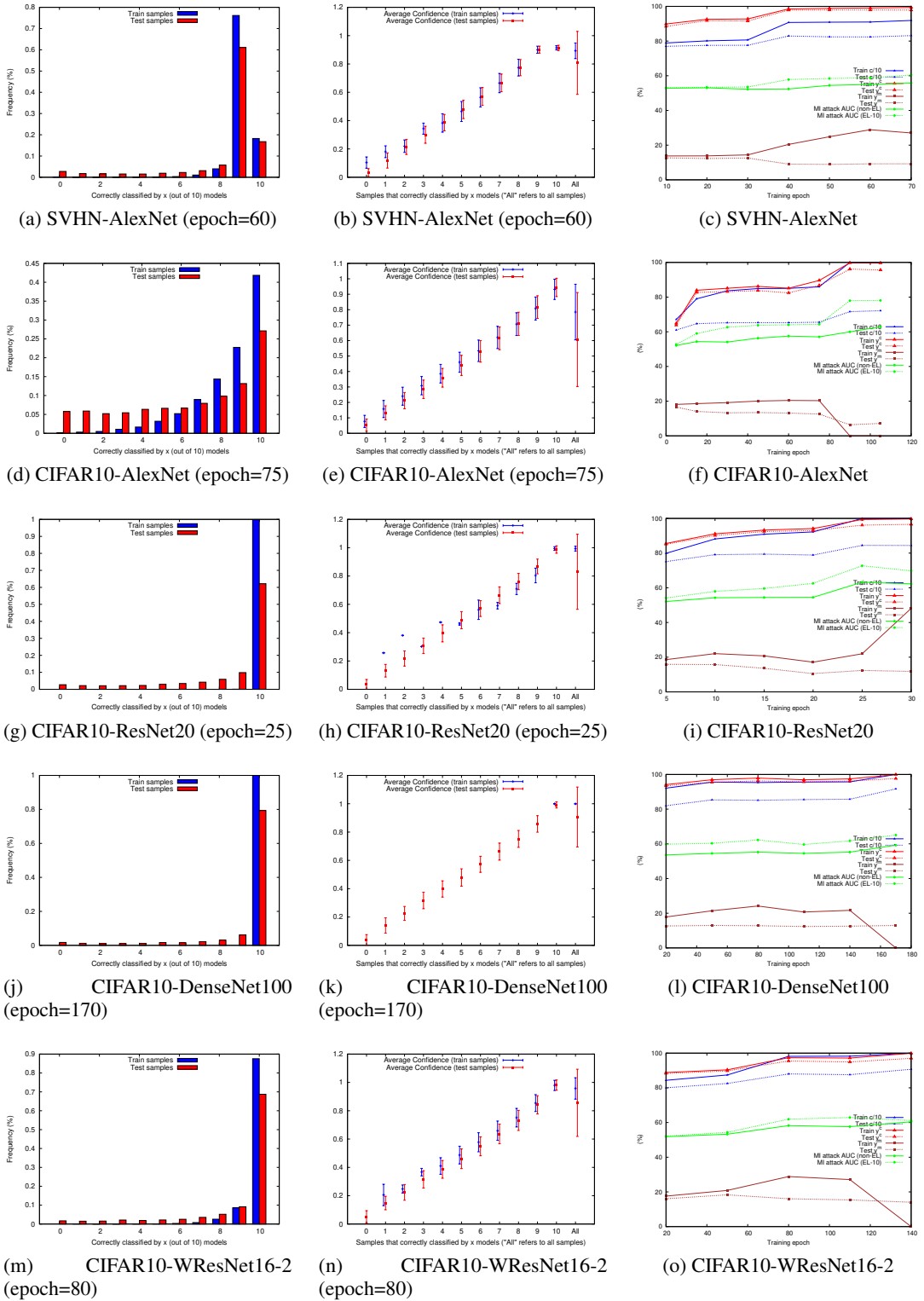

Figure 7: The first row contains correct agreement distribution for SVHN and CIFAR10 datasets. The second row shows the average and standard deviation of distribution of samples based on the level of correct agreement. The third row shows the effect of the three factors on the MI attack.

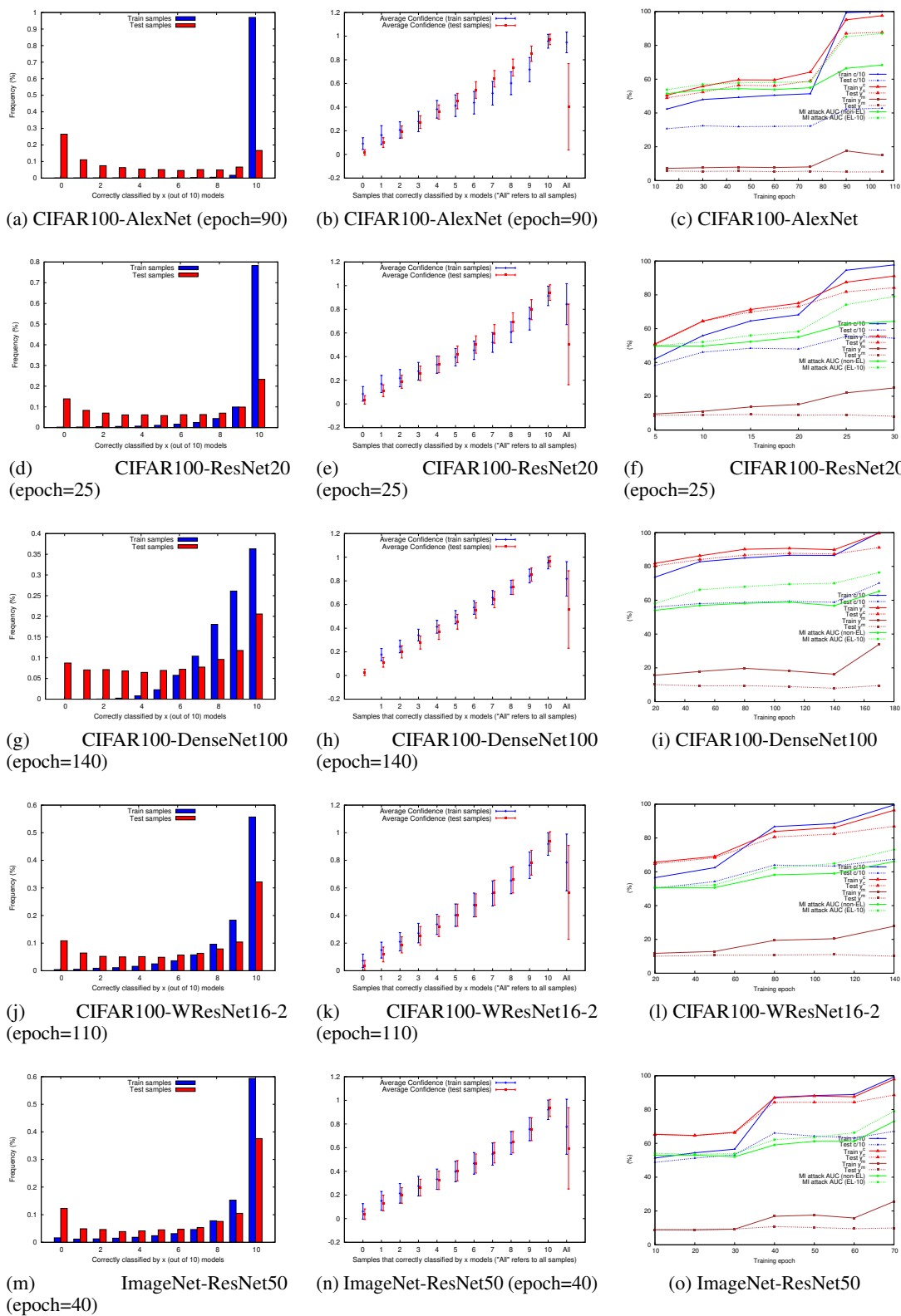

Figure 8: The first row contains correct agreement distribution for CIFAR100 and ImageNet datasets. The second row shows the average and standard deviation of distribution of samples based on the level of correct agreement. The third row shows the effect of the three factors on the MI attack.

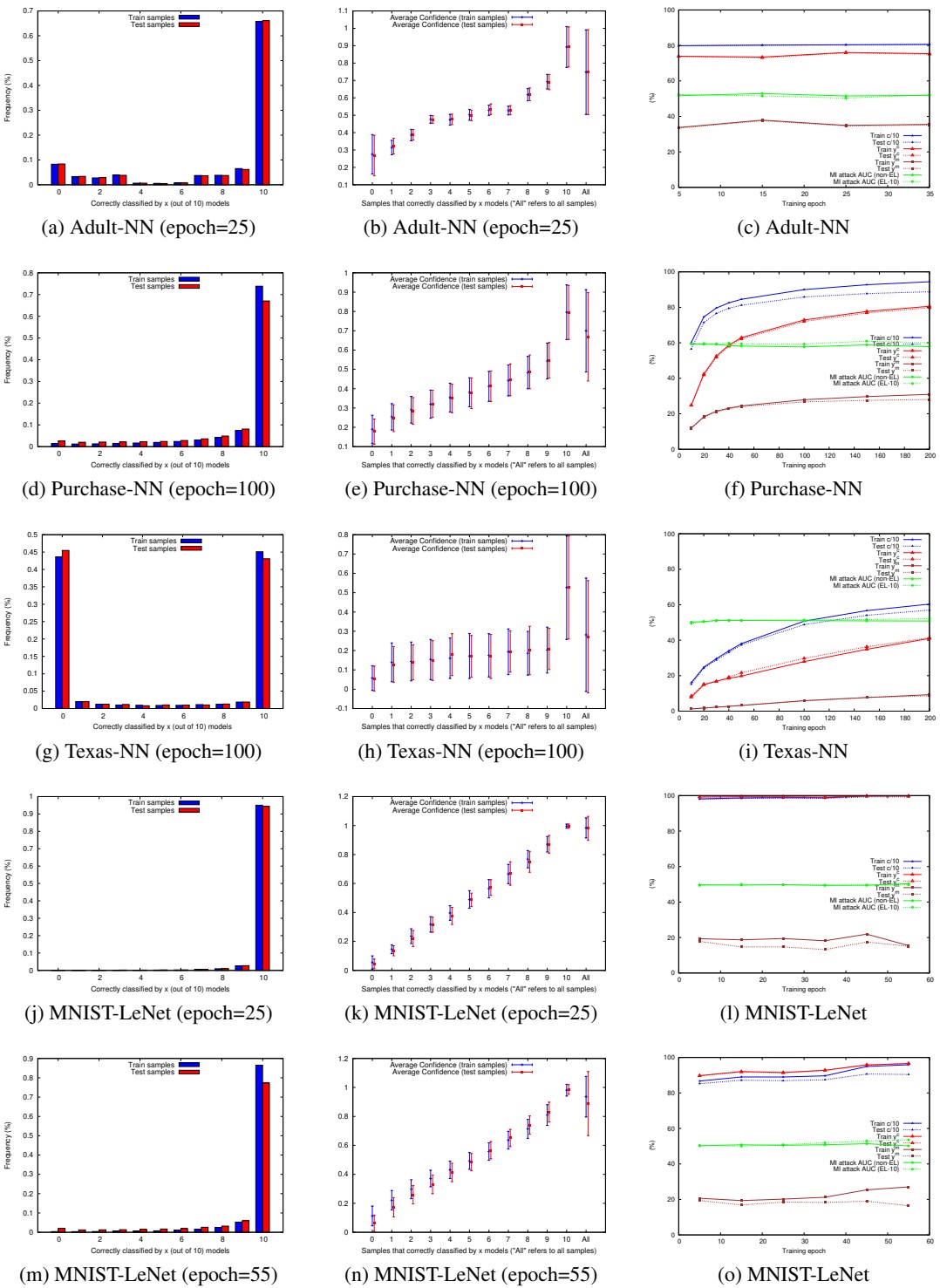

Figure 9: The first row contains correct agreement distribution for Adult, Purchase, Texas, MNIST, and FMNIST datasets. The second row shows the average and standard deviation of distribution of samples based on the level of correct agreement.The third row shows the effect of the three factors on the MI attack.

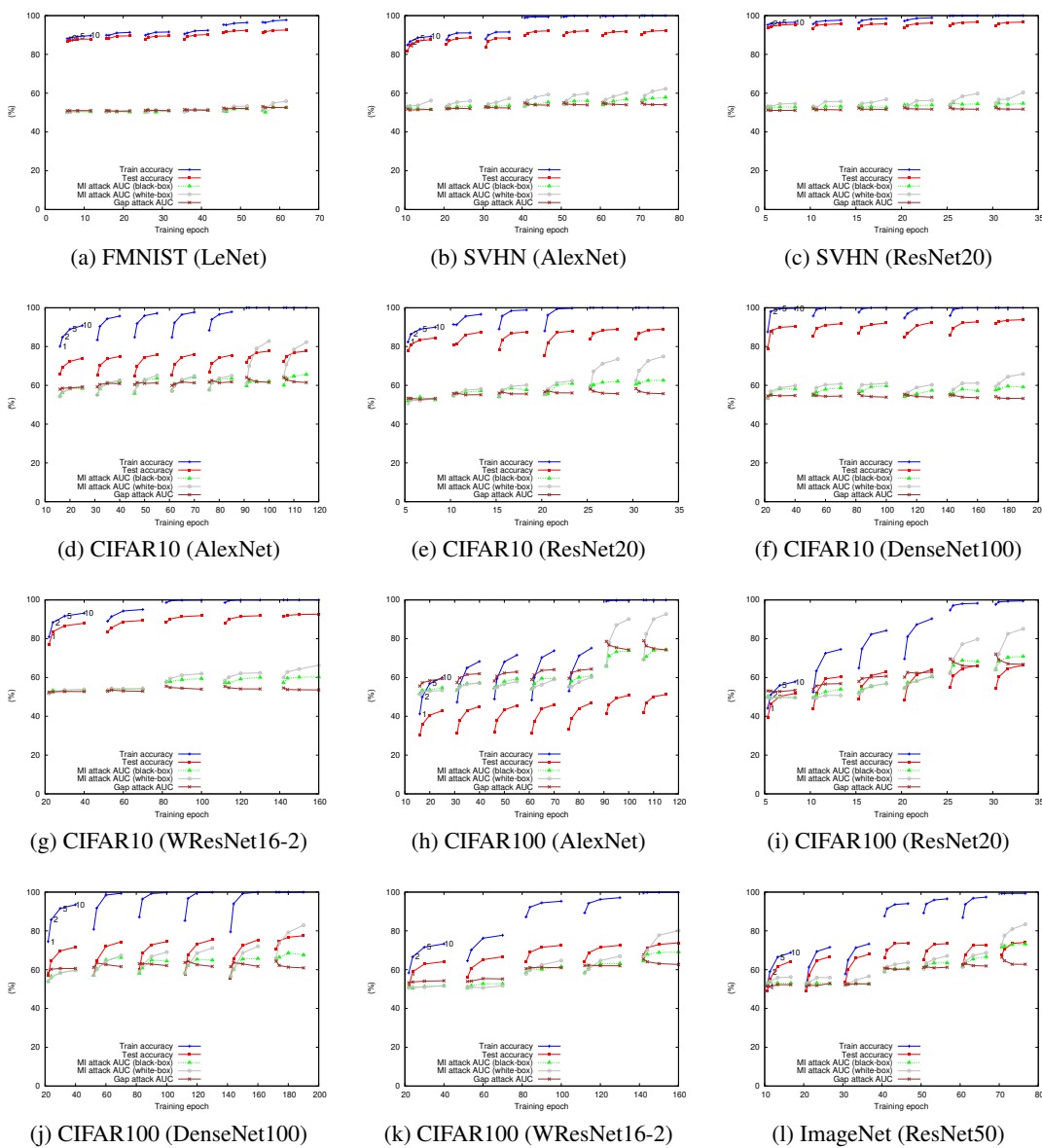

Figure 10: Target models' accuracy and MI attacks' AUC across all datasets and models. Here, MI attack model uses aggregated logits instead of aggregated confidences.

often belong to the correctly classifying models, has significantly more influence on the confidence output of ensemble in comparison with lightly activated neurons of misclassifying models. Hence, the confidence output is heavily influenced by only a portion of models in ensemble that have high activation neurons. In other words, it can be seen as an ensemble of only a portion of models, not all models in the ensemble. Since ensembling with fewer models leaks less membership status, logit averaging of $n$ models leak membership status than confidence averaging of the same number of models. Note that logit averaging is still prone with the same degree to membership leakage in a white-box attack since a white-box attacker has access to all confidence values. Note that the consequence of using logit in certain applications, such as confidence estimation, that requires reliable confidence estimation is out of the scope of this paper. Moreover, one major drawback of using logit is that it can be arbitrary scaled (Wang et al., 2020). However, in scenarios where only accuracy is concerned and white-box access is unavailable to the attacker, averaging logits seems to have a better privacy protection of training data than averaging confidences.

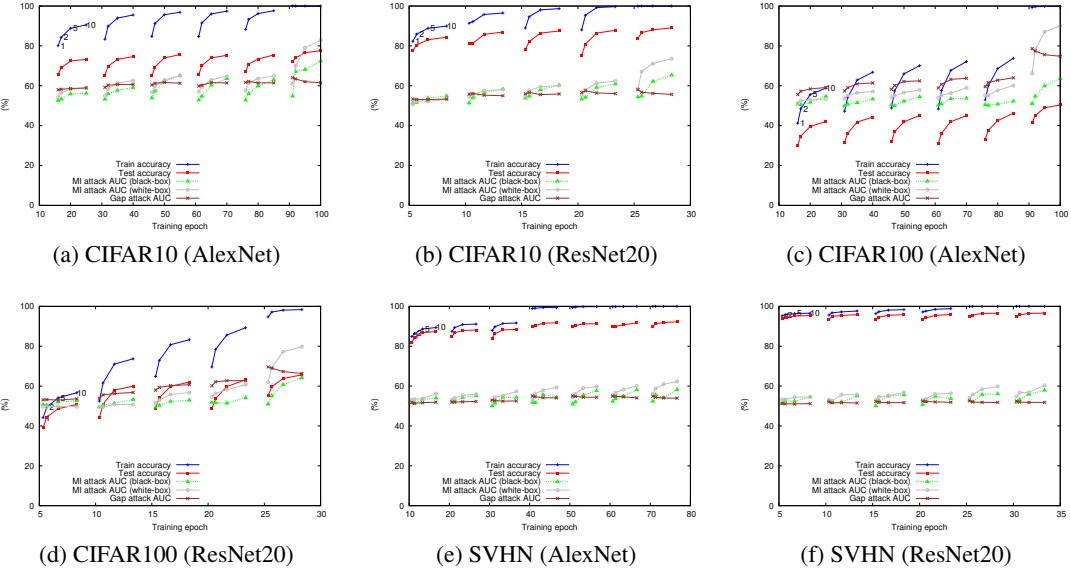

Figure 11: Target models' accuracy and MI attacks' AUC across all datasets and models using weighted averaging ensemble networks.

## A.4 WEIGHTED AVERAGING

In this section, we evaluate weighted averaging of deep models. We focused on image CIFAR10, CIFAR100, and SVHN datasets. We trained each model with random initialization and all hyper-parameters are similar to Section 4.1. Here, we use stochastic gradient descent (SGD) using the entire training set to learn the weight associated with each model. As shown in Figure 11, we observed similar accuracy-privacy trade-off.

## A.5 MORE ADVANCED ENSEMBLING APPROACHES

In this section, we evaluate two state-of-the-art ensembling approaches, namely snapshot ensembles (Huang et al., 2017) and diversified ensemble networks (Zhang et al., 2020). For snapshot ensemble, we train several models on several datasets for 500 epochs and restart the cycle every 50 epochs, similar to the original paper (Huang et al., 2017). Note that the goal of our evaluation is show the accuracy-privacy trade-off, not to achieve the highest accuracy possible. Due to this reason and limited time we had, we did not perform an exhaustive hyper-parameter tuning. Nevertheless, similar trade-off can be observed in Figure 12.

We also conduct the same experiment with diversified ensemble networks (Zhang et al., 2020). The original paper used pre-trained VGG and ResNet models. We did not use pre-trained models for two reasons: 1) It make comparison with other approaches unfair, and 2) It may interfere with the membership inference analysis. We find that by using randomly initialized models to start training, $L_d$ varies significantly and prevents the optimization to converge. Therefore, we add a weight to the $L_d$ term to reduce its effect on the entire loss. We use 0.01 for CIFAR10 and SVHN and 0.001 for CIFAR100. For the shared layer, we use a fully-connected layer of size 128 followed by batch normalization and Relu activation. We use SGD to train models for 60 epochs while dropping the learning at each 20 epoch by 0.1. We could not achieve the exact same results as reported in the paper for two main reasons: 1) we did not use pre-trained models in the ensemble, and 2) many hyper-parameters and implementation details are missed from the original paper. We could not find a set of hyper-parameters and conditions to consistently achieve higher accuracy when increasing the number of models. This was also reported in the original paper where they found that more models in the diversified ensemble do not always improve accuracy. One penitential reason is that training base models in diversified neural networks are not independent. These models are trained during the training of the entire network. This means when the number of models in the diversified neural

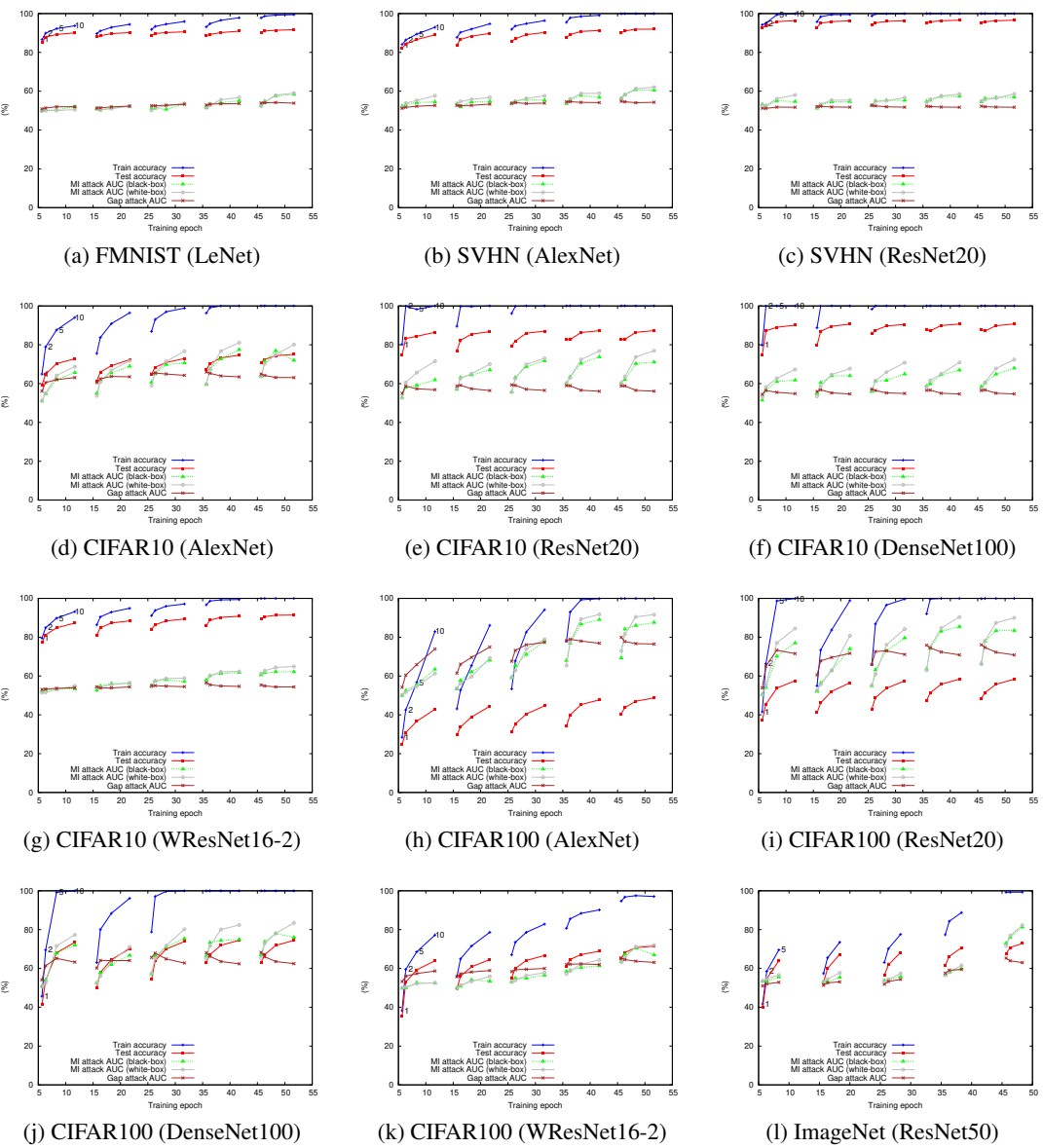

Figure 12: Target models' accuracy and MI attacks' AUC across all datasets and models using snapshot ensemble (Huang et al., 2017).

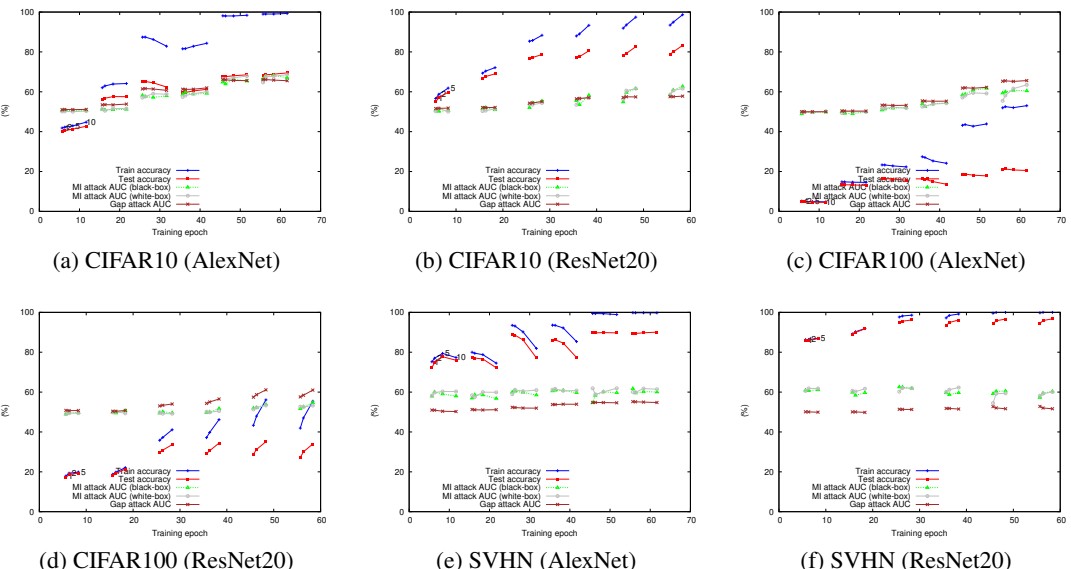

Figure 13: Target models' accuracy and MI attacks' AUC across all datasets and models using diversified ensemble networks (Zhang et al., 2020).

network is increased, there are significantly more parameters to train, but the number of epochs are constant. So, it is expected that diversified neural network with less models to sometimes outperform diversified neural network with more models if the number of training epoch is fixed. Nevertheless, as shown in Figure 13, in cases where accuracy increases, the membership inference attack effectiveness also increases.

