# OpenReview forum: "Accuracy-Privacy Trade-off in Deep Ensemble: A Membership Inference Perspective"
_ICLR.cc/2022/Conference — ICLR 2022 Submitted_

### Official Review · Reviewer_H6K7 · 2021-11-01

**Correctness:** 3
**Technical Novelty And Significance:** 1
**Empirical Novelty And Significance:** 2
**Recommendation:** 6
**Confidence:** 4

**Main Review:**

The authors in the paper perform empirical studies to investigate the trade-off between accuracy and privacy (measured by membership inference attacks) in deep ensembles. They find out that the level of correct agreement among models is the most dominant factor that improves the performance of MI attacks in deep ensembles. They support their claim by visualizing the distribution shifts of correct agreement in train/test examples. They further implement a variety of existing defenses, such as differential privacy and regularizations, etc., to investigate the ​​effects of existing defense mechanisms. Overall, the paper is well-written and the experiments are well conducted.

My biggest concerns on the paper are its limitations of novelties and contributions in the literature of membership inference attacks and privacy-preserving machine learning. The biggest finding of the paper is that the improved advantage of deep ensembles on MI attacks mainly comes from the level of correct agreement among models in the deep ensembles. The finding is purely empirical without theoretical justifications. Besides, it is not sure whether the similar conclusions extend to ensemble learning methods, which limits its contributions to the community.

Besides, the no defense mechanism targeted for the finding is proposed and the defense methods evaluated in the experiments already existed. The privacy-utility trade-offs are expected and it seems the trade-offs are similar to the settings where no deep ensembles are adopted.


**Summary Of The Paper:**

The authors in the paper perform empirical studies to investigate the trade-off between accuracy and privacy (measured by membership inference attacks) in deep ensembles.

**Summary Of The Review:**

Although the paper is well-written and the experiments are well conducted, I think the novelties and contributions of the paper is not enough for acceptance: the authors just find out the level of correct agreement among models in the deep ensembles improves MI attacks in a particular deep ensemble setting, no algorithms nor theoretical insights targeted for the setting is provided.

---

> ### Author Response · Authors · 2021-11-17
> **Response to the reviewer**
>
> We would like to emphasize that the main contribution of the paper is both new and, to some degree counter-intuitive, as far as  previous literature is concerned. In the literature, a specific case of ensembling was proposed by Salem et al. to be used to mitigate MI attack, proposed by Salem et. al,. Then, this was discussed in many other papers as an intuitive solution without conducting any experimental evaluation (Li et al. (2021); Rahimian et al. (2020); Yang et al. (2020)). Looking at the literature, one may conclude that ensemble learning can be used to improve both accuracy and privacy. No where in the privacy or ensembling literature one can find that privacy and accuracy are conflicting goals in ensemble learning. Therefore, to the best of our knowledge, we are the first to identify and analyze the trade-off between accuracy and privacy (in a form of MI attack) in deep ensembles, and ensemble learning, in general.
>
> Furthermore, focusing on deep ensembles,  in Section 3, we provide concrete evidence as to why this happens (that is, why MI attacks are more effective in deep ensembles). Although it is impossible to cover all ensembling methods in a single paper, we cover a number of  state-of-the-art ensembling approaches namely diversified ensemble neural networks and snapshot learning in Appendix A.5, weighted averaging in A.4, and logit averaging in A.3. Then, we show why some ensembling approaches, like bagging, partitioning and stacking, can be used as defense against MI attacks. However, those approaches take a heavy toll on accuracy. Therefore, there is a trade-off between privacy and accuracy.
>
>  Moreover, we show that the overfitting is not the main reason for MI leakage as suggested by literature in deep ensembles. We show that in Figure 4 and in Appendix A.1 that in many cases the generalization gap is decreased when using ensemble and Gap attack effectiveness also decreases. However, MI attack effectiveness actually increases because the confidence values are shifted which is not necessarily correlated with overfitting or generalization gap.
>
> In summary, our findings reported in this paper significantly differ from those in the literature  and they are somewhat counter-intuitive. This is important  because it starts a new challenging research direction where the goal is to make deep ensembles immune to membership inference attacks.

---

> > ### Comment · Reviewer_H6K7 · 2021-11-19
> > **Response after reading the authors' rebuttal**
> >
> > Thanks for your comments and they addresses part of my concerns. I will increase the score by one.

---

### Official Review · Reviewer_G7t1 · 2021-11-01

**Correctness:** 4
**Technical Novelty And Significance:** 2
**Empirical Novelty And Significance:** 3
**Recommendation:** 6
**Confidence:** 4

**Main Review:**

This paper's main insight---that ensembles can be more vulnerable to membership inference---is interesting and a priori somewhat surprising.
But this paper nicely explains how ensembling strengthens privacy leakage: for training examples, all models tend to be correct and highly confident so the ensemble's prediction is essentially the same as any individual model. But for test examples, there is a fraction of examples where the ensemble disagrees. And these become easy to recognize as they have lower confidence.

The most interesting result in my opinion is in Figure 4, where the authors show that as overfitting *decreases* (and thus the trivial gap-attack becomes weaker), membership leakage actually *increases*. I think this result would merit to be discussed more prominently much earlier in the paper, as it runs contrary to some of the beliefs in the literature on membership inference attacks.

While the observation in this paper is quite simple, it is analyzed in depth across a number of setups, datasets, architectures to form a convincing story.
A few suggestions here:
- Figure 1 (& Figure 5) uses a fairly "bad" model that achieves <80% accuracy on CIFAR-10. Do the results still hold if you switch to a better setup (e.g., a Wide-ResNet with standard data augmentation should easily reach >92% accuracy).
- Figure 2 shows that "something" changes qualitatively with the distributions, but it is hard to convincingly say that one should lead to a stronger attack just via visualization. It would be useful here to also provide some quantitative measure of the distinguishability of these distributions (e.g., a simple TV distance).

Other comments:
- The introduction mixes two different forms of ensembling. It first talks about "neural networks (NN) that are independently trained on the same dataset with different random initialization" and then about how "training each model on a different subset of data makes the ensemble less prone to overfitting". This is a bit confusing, since it isn't clear at this point what type of ensembling will be considered in this paper. The note on the use of subsampled ensembles to defend against MI could maybe be relegated to the related work as it is fairly orthogonal to this paper.
- The graphs in the paper are overall a little hard to read, as they tend to show too many things at once. The light green color used in some plots is also hard to see.

**Summary Of The Paper:**

This paper describes and analyzes an interesting and curious phenomenon: a standard ensemble of models is typically more vulnerable to membership inference attacks, despite being less overfit than any single model.

**Summary Of The Review:**

A fairly simple observation but nicely analyzed

---

> ### Author Response · Authors · 2021-11-17
> **Response to the reviewer**
>
> Reviewer concern: The most interesting result in my opinion is in Figure 4, where the authors show that as overfitting decreases (and thus the trivial gap-attack becomes weaker), membership leakage actually increases. I think this result would merit to be discussed more prominently much earlier in the paper, as it runs contrary to some of the beliefs in the literature on membership inference attacks.
>
>
> Our response: The comparison with gap attack and how MI attack in deep ensembles significantly outperform gap attack is indeed an important observation. That is the reason we spend an entire section (Section 3.3) on that. However, the concept of gap attack is not as well-established as other MI attacks in general. Hence, we decided not to focus on that in the introduction to avoid confusion.
>
> -------
>
> Reviewer concern: Figure 1 (& Figure 5) uses a fairly "bad" model that achieves <80% accuracy on CIFAR-10. Do the results still hold if you switch to a better setup (e.g., a Wide-ResNet with standard data augmentation should easily reach >92% accuracy).
>
> Our response: For the main body of the paper, we used models for which ensemble learning improves both training and test accuracy. It was an intentional choice to avoid overestimating MI effectiveness. The reason is because the models that achieve >90% on the test set, often achieve almost perfect accuracy (~100%) on the train set. As a result, there are no misclassified samples or low confidence samples in the training set and, hence, the distribution of the train set after aggregation (in deep ensemble) never changes (Section 3 analysis). However, the distribution of test samples still changes (in fact, decreases). This case causes the most effective MI attack. This is shown in appendix A.1, see Figure 6(h), 6(l), 6(o), etc.
> We also did not use data augmentation because it has been shown that it makes models more susceptible to membership inference attacks (Choo et al.,2020). It would have interfered with our analysis and caused overestimation of MI effectiveness.
> Using any of the suggestions, indeed, would make MI attacks more effective.
>
> --------
>
> Reviewer concern: Figure 2 shows that "something" changes qualitatively with the distributions, but it is hard to convincingly say that one should lead to a stronger attack just via visualization. It would be useful here to also provide some quantitative measure of the distinguishability of these distributions (e.g., a simple TV distance).
>
> Our response: We revised the caption of the figure 2 to provide Jensen–Shannon divergence of the distributions.
>
> --------
>
> Reviewer concern: The introduction mixes two different forms of ensembling. It first talks about "neural networks (NN) that are independently trained on the same dataset with different random initialization" and then about how "training each model on a different subset of data makes the ensemble less prone to overfitting". This is a bit confusing, since it isn't clear at this point what type of ensembling will be considered in this paper. The note on the use of subsampled ensembles to defend against MI could maybe be relegated to the related work as it is fairly orthogonal to this paper.
>
> Our response: We revised Introduction to clarify the difference.

---

> > ### Comment · Reviewer_G7t1 · 2021-11-19
> > **Thanks**
> >
> > Thanks for the response. This addresses my concerns and questions.

---

### Official Review · Reviewer_McDt · 2021-11-02

**Correctness:** 3
**Technical Novelty And Significance:** 2
**Empirical Novelty And Significance:** 2
**Recommendation:** 5
**Confidence:** 4

**Main Review:**

Strength:
1. The idea of studying the accuracy-privacy trade off is interestrting and relatively new.
2. The conclusion on the effectivess of the level of correct agreement among models is important and likely to motivate more related future work towards this direction.
Weakness:
1. The expriments and conclusions are all based on a very simple ensemble setting: averaging the individual predictions.  However in practice, we usually use some more advanced aggregating stratgies such as weighted averaging. In this case, it is not clear whether the conclusions in the paper will still be applicable.
2. Instead of using underfitting ensembles, which usually yield less satisfactory accuracies,  to reduce the MI attack, the authors could also try some advanced ensemble methods. such as those that have enhanced diversities. Some good examples include:
The Diversified Ensemble Neural Network, Deep Negative Correlation Learning


**Summary Of The Paper:**

This paper provide a systemantic analysis on the accuracy-privacy trade off for deep ensmebles. They show that the effectiveness of
membership inference attacks is likely to increase when ensembling improves accuracy. The authors further study  the impact of various factors such as prediction confidence and agreement between models that constitute the ensemble.

**Summary Of The Review:**

Generally speaking, this is an interesting paper because the ``accuraacy and privacy" trade off is under-resaearched. However, the experimental setting is relatively resrective. It is interesting to see whether the conclusion still holds for more advanced and generic ensemble settings.

---

> ### Author Response · Authors · 2021-11-17
> **Response to the comments**
>
> Strength:
> 3 and 4: Since ensemble learning is an umbrella term covering many arbitrary combinations of base models, training procedures, and aggregation mechanisms, it is impossible to cover all of them. Hence, we tried to cover the most common form in detail, namely deep ensemble, and some variations and a few state-of-the-art methods. Due to the lack of space, we mainly move those to the appendix. We covered two state-of-the-art deep ensembling approaches, namely diversified ensemble neural networks and snapshot learning, in Appendix A.5. We covered logit averaging in A.3. We also covered bagging, partitioning and stacking in Section 4.2. We added the results of weighted averaging to the original paper in appendix A.4. We again observed a similar trade-off pattern.

---

> > ### Comment · Reviewer_McDt · 2021-11-25
> > **Thanks for the rebuttal.**
> >
> > Thanks for the rebuttal. My comments have been well-addressed and I would like to vote for an acceptance.

---

### Official Review · Reviewer_YtVz · 2021-11-02

**Correctness:** 3
**Technical Novelty And Significance:** 2
**Empirical Novelty And Significance:** 3
**Recommendation:** 5
**Confidence:** 4

**Main Review:**

## Strengths
1. The division of confidence values by the number of ensemble learners who correctly classify a sample is a very useful way of distinguishing the reason for the distribution change -- it is not because these models have inherently lower confidence on test examples, but because they incorrectly answer them more often. These observations can be very interesting to the whole of MI literature -- because they suggest that all MI attacks in practice do as well as "Gap Attack" (when not considering an ensemble). However, this also makes me wonder if the authors used the strongest attack setting for attacking the ensemble models.
2. The experiments are comprehensively evaluated across a wide variety of datasets, training settings, defense settings -- which helps to be confident about the generality of the results.
3. The ensemble models appear to perform significantly worse in terms of privacy than their base model counterparts. The authors could emphasize the percentage increase in privacy risk.

## Suggestions
1. It would be nice to have some background about previously attributed causes of the success of ensembling techniques. Because from the language in the paper, it appears that the authors are the first to attribute that ensembling increases confidence on train samples, and not on the test set (on average)

2. Since this paper is about "deep ensemble models", it appears incomplete without attention to ensemble-based defenses such as those highlighted by the authors -- like training the ensembles on different subsets of datasets ( which are not necessarily disjoint) -Salem et. al. Or in related work-- Huang et al. (2020); Li et al. (2021); Rahimian et al. (2020); Yang et al. (2020). I would appreciate if this paper was able to categorize scenarios in which ensembles are privacy-preserving, and when they are not.


## Questions
1. Do you have access to all n base models in the ensemble when this is black-box? Based on the graphs, there does not appear to be much difference in the white-box and black-box performance.
2. "We consider a worst-case scenario where 80% of the training dataset is given to the attacker and the goal is to infer the membership of the remaining samples, similar to Rezaei & Liu (2021)" -- Can you explain what this setting is in more detail?
3. Are all models trained on complete training sets? Or on smaller splits of the training dataset (as is done in many MI papers to get the attack performance high)


## Writing
1. Please use \citep for parenthesized citations.
2. For graphs in the appendix, what is the y-axis? It appears to be the reverse of what was followed in the main paper.
4. Sometimes the language of the paper is hand-wavy and unsubstantiated. Such as:
“As they move from under-fitted region to overfitted region, they start to memorize features more specific to train samples Feldman (2020). Consequently, the average gap of correct agreement between train and test set widens. Hence, the wider *the* generalization gap of base learners is, the more effective the membership inference attack would be on *the* ensemble. “
There is no experiment that talks about memorization per se. And membership inference must not be taken as a proxy for memorization to describe itself.
5. “In the black-box setting, the attacker only access the output of the ensemble “ --> has access to

## Post Rebuttal
I am keeping my score unchanged. The following is my response to authors:

---
Thank you for your detailed comments! The paper started with an interesting premise -- some prior work says ensemble methods help reduce MI threat, while there is evidence for ensemble hurting MI. Let us understand what's going on and set things straight. Unfortunately, at the end of the paper, the discussion and experimentation do not live up to this initial expectation. Apart from a small curve for partitioning, bagging in Figure 5, there is hardly any discussion on when and what types of ensemble methods help against MI.

The goal of requesting additional experiments was not to make you do additional experiments and fill up the Appendix, but to highlight how this discussion is very useful for the reader. I thank you for the efforts you put in with the experiments in the Appendix. Unfortunately, they do not see any discussion in the paper. For example, while the authors show how confidence varies with ensemble size -- what happens when this ensemble is not just model randomization but partitioning -- what if this was 10% sets, or 50% sets. What is the tradeoff now? The size of these experiments may seem large, but I am trying to point out from a reader's view -- what types of discussions in the paper (and conclusions) will help resolve the initial dilemma -- should I do ensemble or not? If i want to protect myself from MI.

I am sorry that I am delayed in responding to you. But I believe that while this work is beneficial to the community, it only partially caters to the expectation it built. Re-prioritization and minor additional experiments can make this paper a delight for the reader!


**Summary Of The Paper:**

This work analyzes the accuracy-privacy trade-off in ensemble learning by performing model inference attacks. The key finding of the paper is that the presence of an ensemble (that averages the predictions of individual learners) exacerbates the disparity between the confidence distribution of samples that were seen during training v/s those that weren't. They highlight how the main reason for this observation is the reduced agreement between base models for data points that were not seen during training. There is some evaluation of prior membership inference defenses in the ensemble setting.

**Summary Of The Review:**

While the paper does provide some interesting results about the privacy threat of deep ensemble models, it lacks in the discussion of "deep ensemble models" as a whole -- which has been discussed as a defense in prior work. Further, some of the claims can be further substantiated as highlighted in my review. Finally, it would be good to clarify if the finding of the change in confidence distribution (for deep ensembles) of train and test samples has been observed in prior work, or is a new contribution in this work.

---

> ### Author Response · Authors · 2021-11-17
> **Response to strengths and suggestions**
>
> Strengths
> 1. Response to the concern about whether the authors used the strongest attack setting: First, we need to emphasize that the observation that the confidence-based MI attacks barely outperform Gap attack have been corroborated by multiple studies Rezaei & Liu (2021); Leino & Fredrikson (2020) and it is not our paper’s contribution. We needed to emphasize that this is not the case in ensemble learning anymore. The fact that ensemble learning causes a dramatic shift in confidence distribution that makes MI attacks effective is our contribution. The attack we are using is first studied in Rezaei & Liu (2021) because it gives many advantages to the attacker. In a nutshell, one can consider this MI attack performance as an upper-bound on confidence-based attacks with less advantage. Nevertheless, as we mentioned in the paper, small performance variations of these MI attacks are not the focus in our paper. The reason is that we have shown that the distribution shift causes the member and non-members to be more distinguishable after ensemble. Therefore, any confidence-based MI attack outperforms in ensemble learning in comparison with its non-ensemble counterpart.
>
> Suggestions:
> 1. In the literature, the agreed-upon reason why ensemble learning improves accuracy is the diversity of classification prediction of base learners Kuncheva & Whitaker(2003) and Sagi & Rokach (2018). In summary, a wrong classification by a single model may be corrected when multiple models’ predictions are combined. In such a case, when the goal is to improve model accuracy, the prediction confidence is not of interest and has not been studied. To the best of our knowledge, confidence shift in ensemble learning and its relevance to membership inference has not been studied in neither membership inference literature nor ensemble learning literature.
>
> 2. We covered as many defense mechanisms as possible. Two defense mechanisms based on ensemble learning had been studied in the paper, namely partitioning and bagging. Salem et. al defence mechanism is a stacking ensemble approach where it uses partitioning. The difference is that Salem et. al approach does not average the confidence output of base learning. Instead, it trains another model on the output of base learners. Originally, because it was a variation of partitioning, we did not exactly implement it in the original paper. However, to show that the same results still hold, we add the results of Salem et .al in the updated version of the paper in Figure 5.
> The Huang et al. (2020) paper that you mentioned is a snapshot learning approach. Due to the lack of space, we show the results of that in Appendix A.4. The results of other ensemble approaches are also in the appendix because of the lack of space, such as diversified ensemble networks (A.5) Zhang et al. (2020), logit averaging (A.3), and weighted averaging (A.4).
> Other papers that you have mentioned, which we cited in the Introduction, (Li et al. (2021); Rahimian et al. (2020); Yang et al. (2020)) did not propose any new ensembling-based defense mechanism. They just mentioned the fact that ensemble learning might be helpful as a defense mechanism, based on Salem et. al paper. We revise the second paragraph of Introduction to avoid any confusion.

---

> > ### Comment · Reviewer_YtVz · 2021-11-30
> > **Reply to Author Response**
> >
> > Thank you for your detailed comments! The paper started with an interesting premise -- some prior work says ensemble methods help reduce MI threat, while there is evidence for ensemble hurting MI. Let us understand what's going on and set things straight. Unfortunately, at the end of the paper, the discussion and experimentation do not live up to this initial expectation. Apart from a small curve for partitioning, bagging in Figure 5, there is hardly any discussion on when and what types of ensemble methods help against MI.
> >
> > The goal of requesting additional experiments was not to make you do additional experiments and fill up the Appendix, but to highlight how this discussion is very useful for the reader. I thank you for the efforts you put in with the experiments in the Appendix. Unfortunately, they do not see any discussion in the paper. For example, while the authors show how confidence varies with ensemble size -- what happens when this ensemble is not just model randomization but partitioning -- what if this was 10% sets, or 50% sets. What is the tradeoff now? The size of these experiments may seem large, but I am trying to point out from a reader's view -- what types of discussions in the paper (and conclusions) will help resolve the initial dilemma -- should I do ensemble or not? If i want to protect myself from MI.
> >
> > I am sorry that I am delayed in responding to you. But I believe that while this work is beneficial to the community, it only partially caters to the expectation it built. Re-prioritization and minor additional experiments can make this paper a delight for the reader!

---

> ### Author Response · Authors · 2021-11-17
> **Response to questions and writing**
>
> Questions:
> 1. In black-box setting, the attacker only sees the aggregated confidence, or whatever output the target ensemble approach provides. In white-box setting, the attacker sees and uses the confidence output of all base learners to train the attack model. So, in general, white-box attacks outperform black-box attacks. However, we observe that the difference is not always significant and it is most significant when the number of models in the ensemble is large because then the black-box attack loses more information. Note that our analysis in Section 3 is based on black-box setting because it is more practical and reasonable.
>
> 2. In general, confidence-based MI attack models take the confidence output of a victim model and outputs membership status. To train such an attack model, the attacker needs data with ground truth. Papers such as (Salem et al.(2018); Shokri et al. (2017)) use shadow models to obtain training data for the attack model. They basically use a portion of the training data to train the victim model and the other portion of the training data to train shadow models that give data with ground truth to the attacker. In Rezaei & Liu (2021), authors show that even in the worst-case setting (with respect to the victim), MI attacks are not effective. In their attack, instead of training shadow models to obtain data with ground truth, they assume that the attacker knows some portion of the training and test data of the victim model to train the attack model. We use this attack for several practical reasons: First, for shadow-based attacks (Salem et al.(2018); Shokri et al. (2017)), we need to train 100 shadow models (as suggested in Shokri et al. (2017)) to obtain ground truth data. For our exhaustive evaluation where various combinations of datasets, models, number of base learning and defense mechanisms are examined, it was impractical to train 100 shadow models for each combination. Second, shadow-based attacks need separate training data which is obtained by putting a portion of (half of) original training data aside. Therefore, the victim model is exposed to less training data and is significantly more prone to overfitting. On top of that, in some ensemble approaches, such as bagging and partitioning, each base learner should also be trained on a small portion of an already reduced training data. Hence, the experiment goes to a point where data would not be sufficient to train base learners’ deep models. Third, our main analysis is based on the distribution shift of confidence values. Since we showed that the distribution shift causes the confidence values to be more distinguishable, any confidence-based MI attack is more effective on ensemble learning than on a non-ensemble case, regardless of how the attack training data is obtained. Therefore, we used the simplest MI attack just to quantify the MI effectiveness. We note that  small performance variations of these attack models does not have any impact on the goal of the paper, which is the accuracy-privacy trade-off of deep ensembles.
>
> 3. In the deep ensemble setting, all base models are trained with the entire training set. In Section 4.2, when we evaluated defense mechanisms, we explored bagging and partitioning which require each base model to be trained on a subset of training samples. However, when analysing the MI attack, we assume all training samples are member samples, regardless of the number of base models that were exposed to this sample.
>
> Writing:
> 1, 2 and 4. We appreciate the comments. The updated version is revised accordingly
>
> 3. We revised this entire paragraph of the paper to make sure it is accurate and not out-of-context. We removed the connection with memorization because it was out of context and may cause confusion particularly because we have not defined or elaborated on memorization in the paper. The entire paragraph is as follows in the new version:
>  “Another important observation from Figure 3(c) is that the minimum level of agreement gap between train and test occurs when models are relatively underfitted (i.e., the blue lines in first few epochs).This phenomena has also been partially observed in Fort et al. (2019) (Figure 2(c)). The main reason is that underfitted models often only learn the most common and generalizable features and, thus,they often agree on the features and predictions. As they move from underfitted region to overfitted region, their generalization gaps widen (blue line in Figure 3(c)). As a result, they tend to classify train samples more often than test samples. Consequently, they agree on train samples more than test samples and the average gap of correct agreement between train and test set widens. Hence, the wider generalization gap of base learners is, the more effective membership inference attack would be on deep ensembles.”
>
> Summary Of The Review:
> We revise the paper in both introduction and Section 2 to clarify any confusion.

---

### Decision · Program_Chairs · 2022-01-20

**Decision:**

Reject

**Comment:**

The authors in the paper perform empirical studies to investigate the trade-off between accuracy and privacy (measured by membership inference attacks) in deep ensembles. They find out that the level of correct agreement among models is the most dominant factor that improves the performance of MI attacks in deep ensembles. They support their claim by visualizing the distribution shifts of correct agreement in train/test examples. They further implement a variety of existing defenses, such as differential privacy and regularizations, etc., to investigate the ​​effects of existing defense mechanisms. Overall, the paper is well-written and the experiments are well conducted.
While these findings are interesting, they do not reveal something useful or surprising about deep ensemble learning. It is not clear what the contribution is to the membership inference attack literature and private machine learning literature. they do not propose anything new to make the attacks stronger or defenses stronger.